# Urban vs. Rural Socioeconomic Differences in the Nutritional Quality of Household Packaged Food Purchases by Store Type

**DOI:** 10.3390/ijerph17207637

**Published:** 2020-10-20

**Authors:** Allison Lacko, Shu Wen Ng, Barry Popkin

**Affiliations:** The Carolina Population Center, University of North Carolina at Chapel Hill, Chapel Hill, NC 27516, USA; shuwen@unc.edu (S.W.N.); popkin@unc.edu (B.P.)

**Keywords:** diet quality, nutrition, diet disparities, urban, rural, socioeconomic, income disparities, consumer packaged goods, packaged foods

## Abstract

The U.S. food system is rapidly changing, including the growth of mass merchandisers and dollar stores, which may impact the quality of packaged food purchases (PFPs). Furthermore, diet-related disparities exist by socioeconomic status (SES) and rural residence. We use data from the 2010–2018 Nielsen Homescan Panel to describe the nutritional profiles of PFPs by store type and to assess whether these vary by household urbanicity and SES. Store types include grocery stores, mass merchandisers, club stores, online shopping, dollar stores, and convenience/drug stores. Food and beverage groups contributing the most calories at each store type are estimated using survey-weighted means, while the associations of urbanicity and SES with nutritional quality are estimated using multivariate regression. We find that households that are customers at particular store types purchase the same quality of food regardless of urbanicity or SES. However, we find differences in the quality of foods between store types and that the quantity of calories purchased at each store type varies according to household urbanicity and SES. Rural shoppers tend to shop more at mass merchandisers and dollar stores with less healthful PFPs. We discuss implications for the types of store interventions most relevant for improving the quality of PFPs.

## 1. Introduction

Public health efforts to improve food retail have focused on increased access to grocery stores under the assumption that the food available is healthier than smaller convenience stores [1]. However, research examining trends in household packaged food purchases (PFPs) found that the share of purchases from grocery stores decreased from 2000 to 2012 and that the PFPs that were the top sources of calories for US households did not vary meaningfully by store type from 2000 to 2012 [2]. Further research is needed to determine whether these trends have continued to the present day and whether national trends by store type differ among sociodemographic subpopulations.

Specifically, there may be differences in the nutritional quality of PFPs between urban and rural households. Residents of rural areas have been found to depend more on smaller convenience and dollar stores, which have limited and more expensive food items compared to other store types [3,4] as well as compared to small convenience stores in urban areas [5]. The cost of healthy food in larger grocery stores has also been found to be higher in rural areas [6]. Furthermore, diet-related disparities by socioeconomic status (SES) may be exacerbated in rural versus urban food deserts, where lower income individuals have fewer transportation options (e.g., money for gas or lack of public transportation) to access retail stores with a larger variety of food and/or more affordable prices [7,8]. While many studies focus only on food access and purchases among either the urban poor or rural poor [9], no research exists that studies the intersection of urban/rural residence and SES as it relates to store use and the healthfulness of food purchases. As rural individuals are at higher risk of poorly treated diet-related diseases compared to urban residents [10,11], it is important to understand how the healthfulness of food purchases varies by urbanicity.

This comparative research is urgently needed in a rapidly changing U.S. food system, which includes the growth of mass merchandisers (e.g., Walmart), small dollar stores, and online shopping. For example, dollar stores (Dollar General, Dollar Tree, and Family Dollar) have nearly doubled in the past decade [12]. Their expansion into poor rural towns hurts local grocery stores, often driving them out of business [13,14,15,16]. In poor urban areas that lack traditional grocers, dollar stores tend to cluster, as their small retail footprint allows them to bypass zoning restrictions faced by larger supermarkets [14]. The proliferation of dollar stores in rural and urban low-income neighborhoods makes it especially important to consider the intersection of urbanicity and SES to understand purchasing patterns [17]. Lastly, the COVID-19 outbreak rapidly increased demand for groceries, particularly through online shopping [18]. Therefore, it is important to understand how the quality of online grocery purchases compares to in-store purchases.

PFPs are foods with universal barcodes (e.g., a bag of onions, frozen entrees) and contribute significantly to the healthfulness of the whole diet. Store-bought foods make up most of the U.S. diet [19], and about 70% of calories from store-bought foods come from PFPs (the remainder comes from random weight foods, e.g., loose produce, deli meat) [20]. This has important implications for public health, as packaged foods tend to be highly processed [21] and the intake of highly processed foods is associated with both poor diet quality [21,22] and weight gain [23].

The objectives of this research are as follows: first, determine whether trends in the types of stores households shop at and the nutrient profiles and types of foods/beverages purchased at different store types has changed since 2012; second, investigate whether residence in an urban or rural county is associated with the types of stores shopped at and the nutrient profiles of PFPs within store type; third, understand whether socioeconomic differences in the types of stores shopped at or in the nutritional quality of PFPs varies by urban or rural county of residence. Our analysis uses six store types: grocery stores, mass merchandisers, club stores, online purchases from any store, dollar stores, and other convenience stores. We examine trends beginning in 2010 to provide three years of overlap with prior research (2010–2012).

## 2. Materials and Methods

### 2.1. Data

We used 2010–2018 data from the U.S. Nielsen Homescan Consumer Panel [24,25], where participants track their purchases by scanning food and beverage barcodes and recording the store of purchase. Data from each purchase occasion were aggregated at the annual level for each household to create household-year observations. To be included in Homescan, households must participate for at least ten months each year (*n* = 555,085 household-year observations). We further excluded households if they did not purchase a minimum amount of food and beverages for all quarters in a calendar year or had an incorrect county FIPS code (federal five-digit identifier) (*n* = 3026, 0.5%) for a final analytic sample of 552,059 household-year observations.

Nielsen Homescan is a panel that uses an open cohort study design, where households may exit any time and new households are enrolled to replace dropouts based on demographic and geographic targets. Households in our final sample participated in Nielsen for an average of 4 years. Households are sampled from 52 metropolitan and 24 non-metropolitan markets across the contiguous US and are weighted to be nationally representative. Homescan’s large sample size (about 60,000 households/year) provides a rich demographic and geographic variation of household characteristics which allows for the comparison of urban and rural trends and epidemiological analysis to understand differences by socioeconomic groups.

### 2.2. Store Type

Store type is based on Nielsen’s classification of stores, which is based on store size, annual sales/revenue, and the relative quantity of goods the store carries. Seven non-overlapping categories of stores were analyzed: (1) warehouse clubs (e.g., Costco, Sam’s Club); (2) mass merchandisers–supercenters (e.g., Walmart, Target); (3) grocery stores (e.g., Kroger, Safeway, Whole Foods); (4) dollar stores (e.g., Dollar General, Family Dollar); (5) convenience and drug stores (e.g., CVS, gas stations); (6) online shopping from any store type (e.g., Shoprite.com, Walmart.com); (7) other stores (e.g., non-food retail stores such as Best Buy, liquor stores). Since categories are mutually exclusive, purchases made from a mass merchandiser’s website (e.g., Target.com, Walmart.com) would be categorized as “online shopping” and not as “mass merchandiser”.

### 2.3. Demographic Data

Nielsen provides data on a household’s county of residence, which is updated annually. Following the U.S. Department of Agriculture’s Economic Research Service, counties were categorized as urban or rural using the 2013 Office of Management and Budget “metro” and “nonmetro” delineations [26,27].

Household income was used as a proxy for socioeconomic status. To account for differences in the cost of living across the country, self-reported household income was adjusted using Regional Price Parities from the Bureau of Economic Analysis [28]. Then, income was recalculated as a percent of the Federal Poverty Level (FPL) [29], which accounts for household size, and finally divided into tertiles. Household income tertiles were recalculated each year to reflect changes in household composition and income, Regional Price Parities, and the FPL.

Demographic covariates included education, which was defined as the highest self-reported educational attainment of a head of household and categorized into high school or less, some college, college graduate, or post college graduate; race/ethnicity, which was self-reported for one head of household and was categorized as Hispanic, non-Hispanic (NH) White, NH Black, NH Asian, or NH Other; and household composition, which was based on the self-reported age of each household member and included as a series of count variables for the number of individuals in different age groups (0–1, 2–5, 6–11, 19–64, 65 and older).

### 2.4. Outcomes

Unique barcodes in the Nielsen dataset have been linked to Nutrition Facts Panel data as described elsewhere [30,31]. There is no single measure available to summarize the nutritional quality of packaged foods and beverages (compared to the Healthy Eating Index for overall diet quality [32] or the Grocery Purchase Quality Index, which requires all random-weight and packaged food purchases and excludes mixed dishes [33]). Therefore, a series of outcomes was used to assess the nutritional quality of PFPs. Nutrients of concern included saturated fat, total sugar, and sodium. In addition, we evaluated calories per capita per day from food groups of public health interest (fruits, non-starchy vegetables, processed meats and seafood, mixed dishes, sugar-sweetened beverages (SSBs), and junk foods). The public health relevance for each outcome is detailed in Table 1. In addition, we also examined trends in grains, not as an indicator of nutritional quality but rather because they were a top contributor of calories across store types in prior research [2]. These categories were derived by classifying all products into 27 mutually exclusive food and beverage categories based on Nielsen’s product classifications. Mixed dishes include products such as canned soups and frozen entrees, while junk foods include all salty snacks, grain-based desserts, candy, and sweeteners.

Except for the percent of calories from sugar and saturated fat, all outcomes measured at the household-year level were normalized from annual purchases to daily per capita values. First, we divided annual totals by the number of reliable reporting days that the household participated in the panel. We identified households as “reliable food reporters” if they purchased a minimum amount of food and beverages every three months ($45 for a single-person household and $135 for households with two or more people). Data from reliable reporting quarters within a calendar year were summed to calculate average daily purchases at the household level. Next, daily values were normalized by the number of people in the household in the corresponding year. The proportion of adults and children was later accounted for by adjusting for household composition as a series of covariates.

### 2.5. Statistical Methods

Statistical analysis was conducted using STATA version 15 [42]. To assess trends in purchases from different store types for our first objective, we calculated the percent volume of household PFPs by store type and year by regressing percent volume on the interaction of store type and year. We tested for statistically significant differences between 2010 and 2018 within the same store type. Similarly, to identify the food groups that were the top contributors of calories, we calculated the share of calories for each food group by regressing each food group on the interaction of store type and year. Average values for each year and store type were generated using predictive margins. For top food and beverage groups, we tested for statistically significant differences between stores within the same year, including a global F test for between store comparisons, and between 2010 and 2018 within the same store type.

For our second and third objectives, we used multivariate regression to assess urban and rural differences. We use the standardized nutritional outcomes as detailed in Table 1. For all food group outcomes, a two-part regression was estimated using a probit model and a generalized linear model (GLM) with a gamma distribution and log link. All nutrient outcomes were estimated using only a GLM also with a gamma distribution and log link. To estimate urban and rural differences, STATA’s margins command was used to estimate predicted values by stratifying predictions into urban and rural populations (margins, over(urban)). To assess a potential interaction with socioeconomic status, an interaction term was added between the urbanicity and household income tertile. All regression models were adjusted for income tertile, education, race/ethnicity, household composition, and year.

For all three objectives, we used STATA’s “svy” command to account for survey design (sampling within market strata) and for repeated measurements for those households that participate in the panel for multiple years [43]. While all household-year observations were retained in each model for the correct calculation of standard errors, models were stratified by store type. This was accomplished by using “svy, subpop ():” to limit the analytic subpopulation to those household-years where the household was a reliable reporter and had purchased at least one item from the store type being analyzed.

## 3. Results

Household demographic characteristics by store type are shown in Table 2. Although most households resided in urban counties, Nielsen sampled at least one household from 93.6% of counties in the contiguous U.S. between 2010 and 2018 (90.4% of rural counties and 98.5% of urban counties). Adjusting for the cost of living slightly increased the proportion of low and middle-income households.

### 3.1. Trends in Store Type from 2010 to 2018

We find small but significant changes (*p* < 0.001) between 2010 and 2018 in the volume share of purchases from each store type, except for online shopping. Grocery stores constitute the largest share of volume purchased (57.7% in 2010 and 54.3% in 2018), followed by mass merchandisers (23.2% to 25.6%), club stores (9.3% to 11.0%), convenience and drug stores (3.5% to 2.4%), dollar stores (1.8% to 2.6%), and online shopping (0.7% to 0.8%).

Among households who shopped at each store type, the top food and beverage groups were similar across store types and years (Table 3). Top food groups included grains, salty snacks, desserts, mixed dishes, and candy as the average share of total calories purchased across households. Top beverages included SSBs, plain milk, alcohol, and juice (2010 only). In comparison, fruits and non-starchy vegetables made up a small percent of calories purchased by households. Although the top categories of foods and beverages were the same across store type and time, the relative proportion of calories from each group varied widely by store type (*p*-value < 0.0001 for the F test for each food/beverage group). There were also significant differences between most store pairs using grocery stores as the referent (*p* < 0.0001), with few exceptions (e.g., between grocery stores and mass merchandisers for mixed dishes, SSBs, and alcohol in 2018).

### 3.2. Urban versus Rural Differences

There are statistically significant differences in the daily per capita calories from urban and rural households’ PFPs from different store types, except for grocery stores and convenience/drug stores (Figure 1 and Table 4). Among households that shop at mass merchandisers, rural households purchase almost twice as many calories per person per day as urban households. Among households who shop at dollar stores, rural households also purchase slightly more calories compared to urban households. In comparison, urban households purchase more from club stores.

Although online shopping makes up a small share of volume purchased among all households (<1% across all years), the number of calories purchased among households that did shop online exceeded calories purchased by dollar store shoppers. In 2010, online shoppers purchased an average of 112 calories per person per day while dollar store shoppers purchased 55 calories per person per day. In 2018, this difference narrowed: online shoppers purchased 73 calories per person per day while dollar store shoppers purchased 60 calories per person per day. In addition, rural households purchase more PFPs through online shopping compared to urban households.

To assess differences in quality, results for nutrients and food groups purchased at each store type are summarized in Table 4. Urban–rural differences in the calories purchased from specific foods groups follow the same patterns as total calories purchased. There is little difference in the quality of foods purchased by urban and rural households shopping at the same store type. However, there are differences in the average percent of calories from sugar and from saturated fat purchased by households across store types.

### 3.3. Interaction between Household Income and Urbanicity

Whether there is more variation in calories from PFPs by household income or by urban/rural residence depends on the store type (Figure 2). To assess differences in the nutritional quality of PFPs by income, and whether this further differs by urban/rural residence, results for nutrients and food groups are summarized in Table 5. Store types are condensed into four categories for ease of comparison: online purchases are included with their store type (i.e., purchases from Walmart.com are categorized as “mass merchandisers”) and dollar stores are combined with convenience and drug stores (see Appendix A: Table A1 for all store types).

Differences between high- and low-income households are similar for rural and urban households for all store types except for mass merchandisers. Among households who shop at grocery stores, low-income households purchase more calories from PFPs compared to high-income households. A similar pattern is found among dollar store shoppers. In comparison, among households who shop at club stores, high-income households purchase more calories than low-income households. Among households that shop at mass merchandisers, there is no difference between low-income and high-income rural households.

There are few rural–urban differences in the average calories per person per day purchased from each food group between low- and high-income households. Regardless of urbanicity or store type, high-income households purchase slightly more fruits and non-starchy vegetables. Where statistically significant difference-in-differences are found, the substantive differences are small. As in Table 4, the greatest contrasts can be found in the average percent of calories from sugar purchased by households at specific store types rather than between households categorized by income or urbanicity.

## 4. Discussion

Our study found differences in the nutritional quality of packaged foods and beverages purchased at different types of stores, although overall, quality has changed little from 2010 to 2018 and must be improved. In addition, we found that rural households purchased more calories per person per day from mass merchandisers and dollar stores compared to urban households, and that low-income households purchase more calories from convenience stores and fewer from club stores than high-income households. However, the nutritional quality of those purchases was similar across households shopping at the same store type.

While most packaged foods are obtained from grocery stores, the volume share of purchases has slowly declined, while the share of purchases from mass merchandisers, club stores, and dollar stores has increased. The relative share of volume purchases from 2010 to 2012 are consistent with results from prior research, and the trends we find represent a continuation of trends from 2000 to 2012. [2] Our findings also align with trends in an increasing number of dollar stores in recent years [44]. Our study adds that the share of purchases from other small convenience stores is decreasing. Dollar stores are likely to sustain their growth. If the economic impact of COVID-19 is similar to the Great Recession, dollar stores are likely to see an increase in sales as consumers look for less expensive products [45]. In comparison, while the volume share of purchases from online shopping remained quite low through 2018, COVID-19 has resulted in a rapid increase in online grocery shopping, which is likely to be sustained. This is because shoppers are likely to continue using online shopping once they try it, and companies are likely to accelerate their investment in the infrastructure needed for online shopping to meet demand [45,46]. In addition, by the end of October 2020, 45 states and D.C. had approved new pilot programs to allow online purchases using Supplemental Nutrition Assistance Program (SNAP) in response to COVID-19 [47]. After investing in the infrastructure for online SNAP purchases, these programs are likely to continue.

Top packaged foods and beverages as a share of total calories have not changed meaningfully since 2000. [2] The top food and beverage groups across store types in 2010–2012 were consistent with findings from prior research [2]. Across all years, top groups include a few examples of healthy foods or beverages, apart from unsweetened milk. One exception is that juice has been replaced by alcohol as a top beverage, which is consistent with other findings that juice consumption is declining [48]. Grains are a top contributor of calories and include a mix of healthy and unhealthy grain-based foods. This is notable because it suggests that our unhealthy food groups provide a conservative estimate of low-quality purchases. One recent study compared the nutrient densities of foods to thresholds used by the Chilean government in assigning foods unhealthy warning labels and found that 66% of calories from breads and 100% of calories from ready-to-eat cereals qualified as junk foods by these standards [49]. Therefore, since all breads and cereals were included under grains in our study and grains were the top contributor of calories, our categorization of foods based on Nielsen groupings likely understates the proportion of calories from unhealthy junk foods.

However, while there was little difference in which groups were the top contributors of calories between store types, the relative proportion of calories from these groups differed between stores. The largest difference is in candies, which are not a top contributor of calories from grocery stores or club stores but comprise 10% of calories from mass merchandisers and 28% and 34% of calories from dollar stores and convenience stores, respectively. While the nutritional quality of PFPs is low among all store types, our findings indicate there are important differences between stores. PFPs from dollar stores and convenience stores are particularly high in sugar as a percent of total calories (36% and 39% respectively), and PFPs from mass merchandisers and online shopping (28% and 27%) are slightly higher in sugar than PFPs from grocery stores and club stores (25% and 23%). As a point of reference, the *Dietary Guidelines for Americans* recommend that added sugar not exceed 10% of total caloric intake [50]. While added sugars are a component of our study measure of total sugars, dollar stores and convenience stores are a negligible source of packaged fruits and vegetables with naturally occurring sugars. Relative differences in the percent of calories from sugar in these store types is likely to drive the high numbers of calories from junk foods. Our findings using nutrient outcomes align with previous research using expenditures from Nielsen 2004–2010 to calculate a single healthfulness score of purchases, which found supermarket purchases to be the healthiest, followed by club stores, supercenters (e.g., Walmart), convenience stores, and dollar stores [44]. Store stocking requirements [51] are a potential policy lever to increase the ratio of healthful to unhealthful foods (e.g., eligibility criteria for stores to accept SNAP or local ordinances). However, a recent evaluation of the Minneapolis Staple Food Ordinance found that corporate-owned stores made greater gains in complying with healthy stocking requirements compared to independently owned stores [52]. Therefore, such requirements should be coupled with financing initiatives [53,54] to help independently owned stores stock healthy foods (e.g., refrigerated storage) and community engagement to increase demand for healthy foods to support the commercial viability of such efforts [55].

We find few differences between urban and rural households in the nutritional quality of foods purchased among shoppers at a given store type. However, since meaningful differences do exist in the quality of purchases between stores, the mix of stores that households shop at matters. In particular, rural households tend to shop slightly more at dollar stores and substantially more at mass merchandisers. Therefore, interventions in these store types should be prioritized to reduce urban vs. rural diet-related disparities. Rural households may shop at these stores more frequently due to the lack of grocery stores and/or the convenience of shopping in bulk for food and non-food items at mass merchandisers to save on transportation costs [56,57]. While aspects of food marketing, including the product selection, pricing, point-of-sale promotion, and product placement, influence the healthfulness of purchases (the 4 Ps) [58], further research is needed to understand how these strategies differ between mass merchandisers, dollar stores, grocery stores, and club stores. Since both dollar stores and mass merchandisers consist of only a few large national chains, corporate engagement may be a potential strategy to shift marketing strategies in these stores to improve the quality of PFPs. Advantages of working with national chains include that decisions to shift to healthier products affect many store locations, chains have reliable supplier distribution networks, and chains can negotiate lower prices through buying in bulk [59]. For example, Dollar General aims to equip 5000 of its roughly 15,000 stores with the ability to sell fresh produce, and the Natural Resources Defense Council has secured commitments from Dollar General to prioritize locally sourced produce [60] (product selection). However, evidence suggests that voluntary initiatives by large food retail chains to encourage healthier purchases, through strategies such as reformulation (product selection), reducing prices on healthy foods, and healthy front-of-package labels (promotion) [61], may not be successful by themselves [62] and that corporate stores are more likely to sell and market more unhealthy foods compared to independently owned stores [63]. More research is needed to identify effective healthy retail strategies that are scalable across these large chains and how retailers can be held accountable to voluntary pledges.

We found few differences in the nutritional quality of purchases by the intersection of household income and urban or rural residence. Some patterns in overall shopping emerged. Low-income households purchase more calories from supermarkets, while higher income households purchase more calories from club stores, which is likely due to the membership fees required for these stores. Among households who shop at mass merchandisers, we find in urban counties that low-income households purchase more than high-income households, which is consistent with previous findings [64,65]. However, we find no income difference in purchases from mass merchandisers among rural households. This may be because most barriers faced by rural households, such as the lack of local food stores and long travel time to retailers [66], affect households of all income levels. These findings have implications for the types of retail interventions mass merchandisers should prioritize to promote equity. For example, since rural households purchase far more calories from mass merchandisers compared to urban households but there is no difference in calories purchased by low- vs. high-income rural households, it is important to improve the selection, promotion, and placement of healthy products for all consumers in rural locations. In comparison, mass merchandiser stores located in urban counties should prioritize making healthier PFPs more affordable, since low-income households purchase more than high-income households from these stores.

The use of household purchase data has several limitations. First, the participant burden of scanning each purchase is high, resulting in underreporting, especially for small purchases. However, the degree of measurement error found in Nielsen Homescan is no different than in other household panel datasets [67]. Second, packaged foods are an incomplete picture of household food purchases. Without the inclusion of unpackaged, random-weight purchases (e.g., deli meats, loose produce, bakery goods), we are unable to examine the overall healthfulness of purchases. However, packaged foods make up the majority of foods purchased from stores [19]. Lastly, purchases do not equal consumption. Not all food purchased at the store is consumed, and we are unable to account for food waste. However, the nutritional profile of purchases is correlated with diet quality as measured by 24-h recalls and therefore a good representation of overall intake [68]. Lastly, since Nielsen predominantly samples urban counties, the rural sample is much smaller than our urban sample, and rural households are more likely to be sampled from counties closer to major markets. Therefore, they may not be fully reflective of all rural areas/households.

However, our study has several important strengths. Compared to food retail sales data, household purchase data are directly tied to the sociodemographic characteristics of the household. This allows us to understand how purchasing patterns in urban and rural areas differ among high and low SES households. In comparison, retail sales data can only be linked to area-level measures of socioeconomic characteristics, precluding epidemiological analysis to understand the behavior of different consumer groups. Compared to dietary 24-h recalls, panel data are collected over a longer period of time, which better captures usual intake and avoids bias from seasonality in purchases [30]. In addition, our research group annually updates links between food items and brand-specific Nutrition Facts Panel data, capturing product reformulation and the entry/exit of products to better reflect the nutritional quality of products in a rapidly changing food market. Importantly, our access to data as recent as 2018 allows us to understand how sociodemographic factors are related to the healthfulness of purchasing patterns today.

## 5. Conclusions

While the quality of PFPs must be improved across all store types, we find the quality of PFPs from dollar stores and convenience stores to be worse compared to club stores and supermarkets. Although we find few differences in the quality of foods purchased by urbanicity or household income among shoppers at the same store type, there are significant differences in the types of stores frequented by rural and urban shoppers. Rural shoppers tend to purchase more calories from mass merchandisers and dollar stores with less healthful PFPs. Therefore, interventions should focus on engaging with these chains to offer healthier packaged foods.

## Figures and Tables

**Figure 1 ijerph-17-07637-f001:**
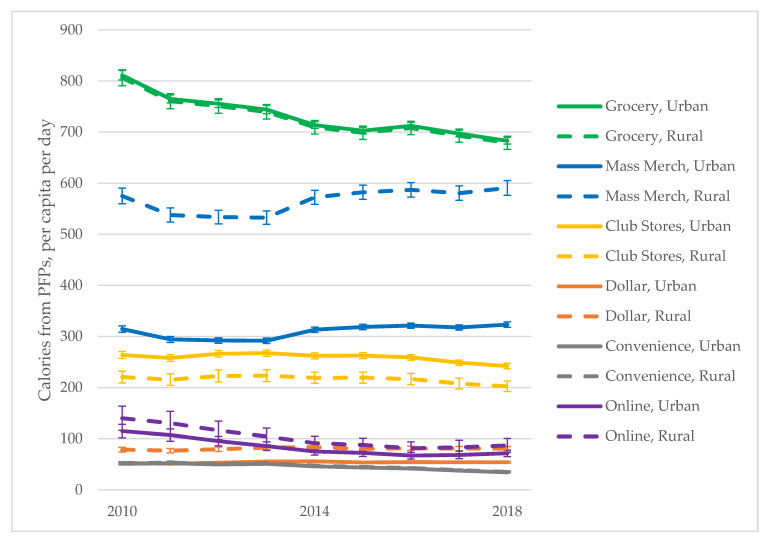
Trends in calories from PFPs by store type and urban/rural household residence. Trends reflect a “per consumer” analysis. Separate models were run for each store type where the analytic sample was limited to those households that purchased at least one packaged food or beverage from a given store type.

**Figure 2 ijerph-17-07637-f002:**
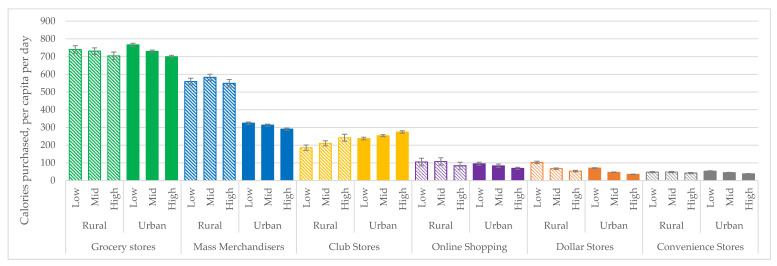
Calories (per person per day) from packaged food purchases, by store type, urban/rural household residence, and household income, 2010–2018.

**Table 1 ijerph-17-07637-t001:** Public health relevance of nutritional outcomes.

NUTRITIONAL OUTCOMES (UNITS)	RATIONALE
Percent of calories from sugar, percent of calories from saturated fat; grams of sugar, grams of saturated fat, mg of sodium (per capita per day)	Overconsumed in the US [34]
	Diets high in sugar are associated with cancer, metabolic syndrome, and obesity [35]
	Replacement of saturated fat with polyunsaturated fat reduces cardiovascular disease risk [36]
	Salt intake associated with cancer [35] and cardiovascular disease [37]
Total calories (per capita per day)	Provide context for calories from select food groups below
Calories from healthy food groups: fruit, non-starchy (NS) vegetables (kcal per capita per day)	Important sources of vitamins and fiber
	High consumption associated with lower cardiovascular disease risk [38]
	Underconsumed in the US [34]
Calories from unhealthy food groups: processed meats, sugar-sweetened beverages (SSBs), junk foods (kcal per capita per day)	Large contributors of total energy, sugar, saturated fat, and sodium in US diet [21]
	The consumption of processed meat is classified as “carcinogenic to humans” by the International Agency for Research on Cancer [39] possibly due to nitrates, higher salt content, and other chemical preservatives [35,40]
	SSBs independently linked to chronic diseases [41]
Calories from grains (kcal per capita per day)	Provide additional context, as grains were the top contributor of calories across store types from 2000 to 2012 [2]

**Table 2 ijerph-17-07637-t002:** Sample characteristics by store type, 2010–2018.

	Club Stores	Mass Merchandisers	Grocery Stores	Online Shopping	Dollar Stores	Convenience Stores	Other Stores
Household-years excluded ^1^	265,050	60,748	12,560	497,906	262,351	185,641	160,444
Analytic Sample	290,035	494,337	542,525	57,179	292,734	369,444	394,641
Demographics: % = Survey-Weighted Proportion (n = Household-Year Observations)							
County of Residence							
Urban	91.0% (261,579)	84.7% (417,186)	85.8% (463,604)	85.5% (48,510)	81.7% (238,636)	86.5% (318,196)	86.3% (338,807)
Rural	9.0% (28,456)	15.3% (77,151)	14.2% (78,921)	14.5% (8669)	18.3% (54,098)	13.5% (51,248)	13.7% (55,834)
Household Income after Adjustment for Cost-of-Living and FPL ^2^							
Low Income (<185% FPL)	20.9% (41,827)	28.1% (97,521)	28.0% (106,065)	29.0% (11,760)	35.5% (71,792)	28.6% (72,673)	26.8% (73,417)
Middle Income (185–400%)	38.5% (122,639)	38.1% (213,942)	37.4% (231,893)	37.2% (24,408)	37.9% (130,612)	37.2% (157,778)	37.1% (166,796)
High Income (>400% FPL)	40.6% (125,569)	33.8% (182,874)	34.6% (204,567)	33.8% (21,011)	26.7% (90,330)	34.2% (138,993)	36.1% (154,428)

^1^ Household-years were excluded from all analyses if the household was a poor food reporter. For both analysis of top contributors of calories and for multivariate regression models, additional household-years were excluded if the household purchased zero packaged food or beverage items from a given store type in that year. Therefore, proportions only include those household-years during which a household shopped at a store type at least once. ^2^ For analysis, household income was adjusted for cost of living, normalized to the Federal Poverty Level (FPL), and then classified into tertiles. Household income is categorized relative to the FPL for ease of comparison in this table. Nielsen disclaimer: Authors’ calculations based in part on data reported by Nielsen through its Homescan Services for all food categories, including beverages and alcohol for the 2008–2018 periods across the U.S. market. The Nielsen Company, 2015. Nielsen is not responsible for and had no role in preparing the results reported herein.

**Table 3 ijerph-17-07637-t003:** Top sources of packaged food purchases (PFP) calories by store type in 2010 and 2018 ^1^ (percent of total calories (SE)).

	Grocery Store	Mass Merchandisers	Club Stores	Online Shopping	Dollar Stores	Convenience Stores	Other Stores	All Stores
Daily calories per capita (SE)								
2010	802 (4.8)	356 (3.5)	257 (3.5)	112 (6.1)	55 (1.3)	51 (0.9)	72 (1.6)	1354 (5.7)
2018	686 (3.5)	358 (2.9)	243 (2.9)	73 (3.5)	60 (1.1)	36 (0.7)	60 (1.2)	1211 (4.5)
Top 5 Food Groups in 2010 (percent of total calories (SE))	Grains 18.1% (0.1%)	Grains 15.1% (0.1%)	Grains 12.3% (0.2%)	Grains 12.5% (0.4%)	Candy 24.5% (0.3%)	Candy 31.0% (0.3%)	Candy 15.4% (0.2%)	Grains 17.1% (0.1%)
	Desserts 8.3% (0.0%)	Candy 11.3% (0.1%)	Salty snacks 10.6% (0.2%)	Candy 12.4% (0.6%)	Desserts 15.6% (0.2%)	Salty snacks 9.0% (0.2%)	Grains 11.4% (0.2%)	Salty snacks 8.6% (0.0%)
	Salty snacks 7.9% (0.0%)	Salty snacks 10.3% (0.1%)	Mixed dishes 8.9% (0.1%)	Salty snacks 9.7% (0.4%)	Salty snacks 15.1% (0.2%)	Desserts 6.7% (0.1%)	Desserts 9.4% (0.2%)	Desserts 7.9% (0.0%)
	Mixed dishes 7.3% (0.0%)	Desserts 10.1% (0.1%)	Desserts 7.8% (0.1%)	Desserts 8.4% (0.4%)	Grains 8.4% (0.2%)	Grains 6.0% (0.1%)	Salty snacks 9.0% (0.2%)	Mixed dishes 7.3% (0.0%)
	Other dairy 6.8% (0.0%)	Mixed dishes 6.7% (0.1%)	Nuts 7.1% (0.1%)	Mixed dishes 6.1% (0.3%)	Mixed dishes 4.2% (0.1%)	Nuts 5.8% (0.1%)	Mixed dishes 3.2% (0.1%)	Fats and oils 6.6% (0.0%)
Top 5 Food Groups in 2018 (percent of total calories (SE))	Grains 16.1% (0.1%) **	Grains 14.3% (0.1%) *	Salty snacks 11.2% (0.1%) **	Grains 13.3% (0.4%)	Candy 27.5% (0.3%)	Candy 33.6% (0.3%)	Candy 16.5% (0.2%) **	Grains 15.3% (0.0%) **
	Salty snacks 8.6% (0.0%) **	Candy 9.9% (0.1%) **	Grains 10.9% (0.1%) **	Salty snacks 10.2% (0.3%)	Desserts 15.0% (0.2%) **	Salty snacks 10.6% (0.2%) **	Salty snacks 10.6% (0.1%) **	Salty snacks 9.1% (0.0%) **
	Other dairy 7.8% (0.0%) **	Desserts 9.8% (0.1%)	Mixed dishes 10.2% (0.1%) **	Candy 9.0% (0.4%)	Salty snacks 11.9% (0.2%)	Desserts 6.4% (0.1%) *	Desserts 9.4% (0.2%)	Desserts 7.6% (0.0%) **
	Desserts 7.8% (0.0%) **	Salty snacks 9.6% (0.1%) **	Desserts 8.0% (0.1%)	Desserts 8.3% (0.3%)	Grains 8.2% (0.1%)	Nuts 5.2% (0.1%) **	Grains 8.2% (0.2%) **	Mixed dishes 7.5% (0.0%) **
	Mixed dishes 7.2% (0.0%) *	Mixed dishes 7.2% (0.1%) **	Nuts 7.0% (0.1%)	Mixed dishes 5.8% (0.3%)	Mixed dishes 4.8% (0.1%) **	Grains 4.8% (0.1%) **	Nuts 2.9% (0.1%)	Other dairy 7.3% (0.0%) **
Top 3 Beverage Groups in 2010 (percent of total calories (SE))	SSBs 5.1% (0.1%)	SSBs 6.5% (0.1%)	SSBs 4.0% (0.1%)	SSBs 6.5% (0.5%)	SSBs 7.3% (0.2%)	SSBs 11.1% (0.2%)	Alcohol 18.9% (0.3%)	SSBs 5.1% (0.0%)
	Milk 4.8% (0.0%)	Milk 3.4% (0.1%)	Milk 2.5% (0.1%)	Milk 3.9% (0.3%)	Milk 1.2% (0.1%)	Milk 7.7% (0.2%)	SSBs 6.3% (0.2%)	Milk 4.2% (0.0%)
	Juice 1.9% (0.0%)	Juice 1.7% (0.0%)	Juice 2.1% (0.1%)	Alcohol 2.4% (0.3%)	Juice 0.9% (0.1%)	Alcohol 5.1% (0.2%)	Milk 1.7% (0.1%)	Alcohol 2.1% (0.0%)
Top 3 Beverage Groups in 2018 (percent of total calories (SE))	SSBs 4.3% (0.0%) **	SSBs 4.5% (0.1%) **	SSBs 2.7% (0.1%) **	SSBs 5.4% (0.3%)	SSBs 7.7% (0.1%)	SSBs 11.6% (0.2%)	Alcohol 21.2% (0.3%) **	SSBs 4.1% (0.0%) **
	Milk 3.8% (0.0%) **	Milk 3.1% (0.0%) **	Alcohol 2.3% (0.1%) **	Milk 2.6% (0.2%) *	Milk 1.9% (0.1%) **	Alcohol 5.8% (0.1%) **	SSBs 5.8% (0.1%) *	Milk 3.5% (0.0%) **
	Alcohol 1.9% (0.0%) **	Alcohol 1.7% (0.0%) **	Milk 2.1% (0.1%) **	Alcohol 1.8% (0.2%)	Juice 0.9% (0.0%)	Milk 4.9% (0.1%) **	Milk 1.1% (0.0%) **	Alcohol 2.4% (0.0%) **
Other groups, 2010 (percent of total calories (SE))	Fruits 1.4% (0.0%)	Fruits 1.3% (0.0%)	Fruits 4.1% (0.1%)	Fruits 2.5% (0.2%)	Fruits 1.4% (0.1%)	Fruits 1.2% (0.1%)	Fruits 1.2% (0.0%)	Fruits 1.6% (0.0%)
	Vegetables 1.5% (0.0%)	Vegetables 0.8% (0.0%)	Vegetables 1.9% (0.1%)	Vegetables 1.3% (0.2%)	Vegetables 1.1% (0.1%)	Vegetables 0.4% (0.0%)	Vegetables 0.8% (0.0%)	Vegetables 1.2% (0.0%)
Other groups, 2018 (percent of total calories (SE))	Fruits 1.8% (0.0%) **	Fruits 1.7% (0.0%) **	Fruits 4.3% (0.1%) *	Fruits 2.3% (0.2%)	Fruits 1.1% (0.0%) **	Fruits 0.7% (0.0%) **	Fruits 1.4% (0.1%) *	Fruits 1.9% (0.0%) **
	Vegetables 1.9% (0.0%) **	Vegetables 1.3% (0.0%) **	Vegetables 2.1% (0.1%) *	Vegetables 1.7% (0.1%)	Vegetables 1.0% (0.1%)	Vegetables 0.4% (0.0%)	Vegetables 0.9% (0.0%)	Vegetables 1.6% (0.0%) **

^1^ In 2010, the sample size across all store types was 61,105 household-year observations. In 2018, the sample size was 61,372 household-year observations. For each household-year observation, calories were summed for each food and beverage group as well as across all purchases to calculate a household’s share of calories from each group for a given year. Results represent the average share of calories from each food/beverage group across households that purchased at least one PFP from a given store type in a given year. * 2010 vs. 2018 difference significant at *p* < 0.05 for share of calories purchased from food group within store type. ** 2010 vs. 2018 difference significant at *p* < 0.001 for share of calories purchased from food group within store type. SSB: Sugar-sweetened beverage. Nielsen disclaimer: Authors’ calculations based in part on data reported by Nielsen through its Homescan Services for all food categories, including beverages and alcohol for the 2008–2018 periods across the U.S. market. The Nielsen Company, 2015. Nielsen is not responsible for and had no role in preparing the results reported herein.

**Table 4 ijerph-17-07637-t004:** Urban vs. rural nutritional quality of household packaged food purchases by store type, 2010–2018 ^1^ (predicted mean (SE)).

	Grocery Stores			Mass Merchandisers			Club Stores			Online Shopping			Dollar Stores			Convenience/Drug		
	Total	Rural	Urban	Total	Rural	Urban	Total	Rural	Urban	Total	Rural	Urban	Total	Rural	Urban	Total	Rural	Urban
Total calories, person/day ^2^	730 (2)	728 (6)	731 (2)	349 (2)	565 (6) **	310 (2)	255 (2)	216 (5)	258 (2) **	85 (2)	100 (7)	82 (3)	59 (1)	81 (2) **	54 (1)	46 (1)	47 (1)	45 (1)
Fruits	10 (0)	9 (0)	11 (0) **	5 (0)	7 (0) **	4 (0)	10 (0)	7 (0)	10 (0) **	1 (0)	2 (0)	1 (0)	1 (0)	1 (0)	1 (0)	0 (0)	0 (0)	0 (0)
NS Vegetables	11 (0)	10 (0)	11 (0) **	4 (0)	6 (0) **	3 (0)	4 (0)	3 (0)	4 (0) **	1 (0)	1 (0)	1 (0)	0 (0)	0 (0)	0 (0)	0 (0)	0 (0)	0 (0)
Processed meats	33 (0)	39 (1) **	32 (0)	14 (0)	25 (0) **	12 (0)	10 (0)	9 (0)	10 (0) **	3 (0)	4 (1)	3 (0)	1 (0)	2 (0) **	1 (0)	0 (0)	1 (0)	0 (0)
Mixed dishes	53 (0)	49 (1)	54 (0) **	28 (0)	43 (1) **	25 (0)	21 (0)	15 (1)	22 (0) **	6 (0)	6 (1)	6 (0)	3 (0)	4 (0) **	3 (0)	1 (0)	1 (0)	1 (0)
Grains	124 (1)	119 (1)	125 (1) **	55 (0)	88 (1) **	49 (0)	33 (0)	26 (1)	34 (0) **	14 (1)	16 (1)	13 (1)	7 (0)	9 (0) **	6 (0)	3 (0)	4 (0)	3 (0)
SSBs	33 (0)	35 (1) **	33 (0)	17 (0)	28 (1) **	15 (0)	8 (0)	6 (0)	8 (0) **	4 (0)	4 (0)	4 (0)	4 (0)	7 (0) **	4 (0)	5 (0)	6 (0) **	4 (0)
Junk foods	170 (1)	173 (2)	169 (1)	103 (1)	167 (2) **	92 (1)	66 (1)	62 (2)	67 (1)	23 (1)	29 (2) **	22 (1)	28 (0)	38 (1) **	26 (0)	20 (0)	18 (1)	20 (0) **
Sugar g	45 (0)	45 (0)	45 (0)	23 (0)	37 (0) **	20 (0)	14 (0)	12 (0)	14 (0) **	5 (0)	6 (0) **	5 (0)	5 (0)	7 (0) **	4 (0)	4 (0)	4 (0)	4 (0)
Sugar, % Total calories	25% (0.0%)	25% (0.1%) **	24% (0.0%)	28% (0.0%)	28% (0.1%)	28% (0.0%) **	23% (0.1%)	24% (0.2%) **	23% (0.1%)	27% (0.2%)	28% (0.4%) **	27% (0.2%)	36% (0.1%)	37% (0.2%)	36% (0.1%)	39% (0.1%)	40% (0.2%) **	38% (0.1%)
Sat fat g	10 (0)	11 (0) **	10 (0)	5 (0)	8 (0) **	4 (0)	4 (0)	3 (0)	4 (0) **	1 (0)	1 (0)	1 (0)	1 (0)	1 (0) **	1 (0)	1 (0)	1 (0)	1 (0)
Sat fat, % Total calories	13% (0.0%)	13% (0.0%) **	13% (0.0%)	12% (0.0%)	12% (0.0%) **	12% (0.0%)	13% (0.0%)	12% (0.1%)	13% (0.0%)	11% (0.1%)	11% (0.2%)	11% (0.1%)	10% (0.0%)	10% (0.1%) **	10% (0.0%)	12% (0.0%)	12% (0.1%)	12% (0.0%) **
Sodium mg	1393 (5)	1412 (14)	1390 (5)	687 (4)	1123 (13) **	608 (4)	474 (5)	425 (11)	479 (5) **	172 (6)	204 (15)	167 (6)	148 (2)	184 (5) **	140 (2)	57 (1)	63 (2) **	57 (1)

^1^ The estimates presented are based on a “per consumer” analysis, where the analytic sample for each store type was limited to households purchasing at least one item from a given store type in a year. In other words, an average of 733 calories per person per day are purchased from grocery stores among households who purchased at least one PFP from a grocery store. Estimates are generated using multivariate regression, controlling for household income, education, race/ethnicity, age composition, and year and are adjusted for Nielsen’s survey design to be nationally representative. ^2^ Total calories and calories from specific food groups are expressed in units of calories per person per day. Grams of sugar and saturated fat are expressed in grams purchased per person per day. Percent purchased calories from sugar or saturated fat are calculated by dividing the calories attributable to sugar/saturated fat across all households by the total number of calories purchased from that store type across all households. Mixed dishes include foods such as frozen entrees and canned soups, and junk foods include salty snacks, grain-based desserts, candy, and sweeteners. Non-starchy (NS) vegetables include leafy greens but not potatoes, corn, etc. Examples of mixed dishes include frozen entrees and canned soups; examples of junk foods include candies and salty snacks. SSBs: sugar-sweetened beverages. ** Urban or rural value that is statistically higher at a significance value of *p* < 0.01. Nielsen disclaimer: Authors’ calculations based in part on data reported by Nielsen through its Homescan Services for all food categories, including beverages and alcohol for the 2008–2018 periods across the U.S. market. The Nielsen Company, 2015. Nielsen is not responsible for and had no role in preparing the results reported herein.

**Table 5 ijerph-17-07637-t005:** Urban/Rural comparison of differences by income in the nutritional quality of household packaged food purchases by store type, 2010–2018 ^1^ (predicted margin (SE)).

Store Type	Grocery Stores				Mass Merchandisers				Club Stores				Convenience Stores			
County Type	Rural		Urban		Rural		Urban		Rural		Urban		Rural		Urban	
Income Tertile	Low	High	Low	High	Low	High	Low	High	Low	High	Low	High	Low	High	Low	High
Total calories, person/day ^2^	740 (10.6) *	704 (10.9)	770 (4.6) **	702 (3.8)	564 (9.5)	555 (10.8)	329 (3.3) **	294 (3.2)	188 (7.6)	245 (10.0) **	242 (3.9)	281 (3.5) **	122 (3.4) **	74 (2.5)	94 (1.4) **	53 (0.9)
Fruits	9 (0.2)	10 (0.3) **	10 (0.1)	11 (0.1) **	6 (0.2)	8 (0.2) **	4 (0.1)	4 (0.1) *	5 (0.3)	8 (0.6) **	8 (0.2)	11 (0.2) **	1 (0.0) **	1 (0.1) **	1 (0.0)	1 (0.0)
NS Vegetables	9 (0.2)	11 (0.2) **	11 (0.1)	12 (0.1) **	5 (0.1)	6 (0.2) *	3 (0.0)	4 (0.1) **	2 (0.2)	4 (0.3) **	4 (0.1)	5 (0.1) **	0 (0.0) **	0 (0.0)	1 (0.0) **	0 (0.0)
Processed meats	41 (0.8) **	35 (0.9)	34 (0.3) **	30 (0.3)	25 (0.6) *	23 (0.6)	13 (0.2) **	11 (0.2)	8 (0.5)	10 (0.8) *	9 (0.2)	11 (0.2) **	2 (0.1) **	1 (0.1)	2 (0.1) **	1 (0.0)
Mixed dishes	53 (1.1) **	44 (1.0)	60 (0.6) **	48 (0.4)	46 (1.4) **	39 (1.2)	28 (0.4) **	22 (0.3)	13 (0.7)	17 (0.9) *	21 (0.4)	23 (0.4) *	6 (0.3) **	2 (0.2)	5 (0.1) **	2 (0.1)
Grains	121 (2.1) *	114 (2.0)	131 (1.0) **	120 (0.8)	87 (1.7)	85 (2.0)	52 (0.6) **	46 (0.5)	23 (1.2)	28 (1.8) *	32 (0.8)	34 (0.6)	13 (0.5) **	7 (0.3)	10 (0.2) **	5 (0.1)
SSBs	38 (1.1) **	30 (1.0)	39 (0.5) **	27 (0.4)	32 (1.0) **	24 (0.8)	17 (0.3) **	12 (0.2)	6 (0.4)	6 (0.6)	9 (0.3) *	8 (0.2)	13 (0.6) **	7 (0.4)	8 (0.2) **	4 (0.1)
Junk Foods	172 (2.6)	172 (3.2)	174 (1.4) **	166 (1.1)	162 (3.0)	169 (3.4)	95 (1.0) *	90 (1.0)	53 (2.3)	72 (3.4) **	61 (1.1)	74 (1.0) **	52 (1.4) **	34 (1.3)	42 (0.7) **	2 6 (0.5)
Sugar, g	47 (0.7) *	43 (0.8)	48 (0.3) **	42 (0.3)	38 (0.7) *	36 (0.7)	22 (0.2) **	19 (0.2)	11 (0.6)	14 (0.6) *	14 (0.3)	15 (0.2) **	11 (0.3) **	7 (0.2)	8 (0.1) **	5 (0.1)
Sugar, % Total calories	25% (0.1%) **	25% (0.2%)	25% (0.1%) **	24% (0.1%)	28% (0.2%) *	28% (0.2%)	29% (0.1%) **	28% (0.1%)	25% (0.5%) *	23% (0.3%)	24% (0.2%) **	23% (0.1%)	37% (0.3%)	37% (0.3%)	37% (0.1%)	37% (0.1%) *
Saturated fat, g	11 (0.2)	10 (0.2)	11 (0.1) **	10 (0.1)	8 (0.1)	8 (0.2)	5 (0.0) **	4 (0.0)	3 (0.1)	4 (0.2) **	4 (0.1)	4 (0.1) **	1 (0.0) **	1 (0.0)	1 (0.0) **	1 (0.0)
Saturated fat, % calories	13% (0.1%)	13% (0.1%) *	12% (0.0%)	13% (0.0%) *	12% (0.1%)	12% (0.1%)	12% (0.0%)	12% (0.0%)	12% (0.2%)	13% (0.1%)	13% (0.1%)	13% (0.1%) *	11% (0.1%)	11% (0.1%) **	11% (0.0%)	12% (0.1%) **
Sodium, mg	1441 (21.9) *	1365 (23.4)	1468 (10.0) **	1346 (8.6)	1130 (20.6)	1089 (22.6)	652 (7.7) **	577 (7.7)	378 (15.3)	480 (19.0) **	472 (9.3)	512 (7.4) *	251 (8.4) **	131 (5.4)	182 (3.2) **	93 (2.1)

^1^ The estimates presented are based on a “per consumer” analysis, where the analytic sample for each store type was limited to households purchasing at least one item from a given store type in a year. Estimates are generated using multivariate regression, controlling for household income, education, race/ethnicity, age composition, and year and are adjusted for Nielsen’s survey design to be nationally representative. Store types are condensed into four categories for ease of comparison: online purchases are included with their store type (i.e., purchases from Walmart.com are categorized as “mass merchandisers”) and dollar stores are combined with convenience and drug stores (see Table A1 in Appendix A for all store types). ^2^ Total calories and calories from specific food groups are expressed in units of calories per person per day. Grams of sugar and saturated fat are expressed in grams purchased per person per day. Percent purchased calories from sugar or saturated fat are calculated by dividing the calories attributable to sugar/saturated fat across all households by the total number of calories purchased from that store type across all households. Non-starchy (NS) vegetables include leafy greens but not potatoes, corn, etc. Mixed dishes include foods such as frozen entrees and canned soups, and junk foods include salty snacks, grain-based desserts, candy, and sweeteners. SSBs: sugar-sweetened beverages. * Indicates a statistically significant difference between low-income households and high-income households at *p* < 0.05. ** Indicates a statistically significant difference between low-income households and high-income households at *p* < 0.001. The shaded cells indicate that the income difference for rural (urban) households is significantly greater than for urban (rural) households—in other words, the difference-in-difference is significant, *p* < 0.05. Nielsen disclaimer: Authors’ calculation based in part on data reported by Nielsen through its Homescan Services for all food categories, including beverages and alcohol for the 2008–2018 periods across the U.S. market. The Nielsen Company, 2015. Nielsen is not responsible for and had no role in preparing the results reported herein.

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
