# Peer review of "Urban vs. Rural Socioeconomic Differences in the Nutritional Quality of Household Packaged Food Purchases by Store Type"

_ijerph, 2020, doi:10.3390/ijerph17207637_

Round 1

Reviewer 1 Report

This study is very interesting and I have learned something from it. Thank you for the hard work and providing this information. Here are my comments after reading your manuscript:

  1. It sometimes confused me when reading the texts where it says “households” but in the table/figure the values are presented as “per person per day”. When I see “households” I consider it the whole family. While it makes sense that these values are presented as per person per day, and it is good that the authors are referring it as so, I would suggest making it more clear by using a unified term or add an explanation in the texts. 
  2. Suggest capitalizing COVID-19 since it is the abbreviation of Coronavirus Disease 2019.  

Reviewer 2 Report

The aim of the study was to assess Urban vs Rural Socioeconomic Differences in the Nutritional Quality of Household Packaged Food 3 Purchases by Store Type.

Methodology is complex and not clear – there are many weaknesses, described in limitation section that It would be useful to compare individual behaviours in terms of both food purchases and food intake, as well as the differences in mentioned variables. How many of rural respondents have a possibility to grow their own food? The access to own production may be crucial in purchases need for rural inhabitants, especially when taking into consideration total caloric intake analyses.

The manuscript has been submitted to IJERPH while the Nutrients logo is used in the file.

Line 68: is the first aim of the study correctly described? Authors mentioned “changed since 2012”, while through the manuscript there is a date of 2010 included (i.e. line: 76, 143, 165).

Line 76-78: a reference is needed

Line 180: “Table 2A” suggests that there is also another Table 2 (Table 2B?) – could author clarify it?

Line 216: there is reference to Appendix A: Table A1 included, however there is no appendix in enclosed files.

Reviewer 3 Report

Thank you for the opportunity to review this manuscript on trends in packaged food purchases according to store type, rurality and income level. This study is generally well-written and provides new insights on population purchasing behaviours. Below I outline a several aspects of the manuscript that could be further clarified to improve the reporting of the study design and outcomes.

Introduction:

Lines 66-68: The stated aim of this study is to compare trends from 2012 to 2018 yet the analyses compare 2010 to 2018. It is unclear why these timepoints do not align?

Methods:

Lines 76-80: The study design is not clear. Whilst Homescan is a longitudinal cohort, were the data analysed cross-sectionally or as a cohort study? Throughout the manuscript, it is not always clear whether pooled or annual household-year observations were analysed. The unit ‘household-year observations’ should be defined to add clarity. How was it calculated?

Line 76: Is this the US Homescan panel? The country and setting of this study could be additionally specified.

Line 78: should this refer to "recording the store of purchase for at least ten months PER YEAR/OVER THE 8 YEAR PERIOD"?

Line 83: Should this sentence read as “Unique barcodes IN THE NIELSEN DATASET have been linked to…”?

Lines 99-104: Was household income equivalised to reflect household composition?

Lines 112-135: The units of analysis are not reported consistently throughout the manuscript, particularly in the Outcomes, Statistical Analysis and Results sections (e.g. calories per capita per day, percent volume of household PFPs, daily values, (nutritional) quality, top contributors of calories per household? Per capita?). Are outcomes annual averages?

The Outcomes section could more clearly list the number and types of outcomes measured.

In the Statistical Analysis section, it is also important to more clearly convey which outcomes were estimated at the household vs. individual level. This can help to determine whether the correct analysis methods were used given how mixed regression models are often relevant to account for clustered purchasing behaviours at the household level.

Lines 112-135: It is not apparent why the food and nutrient profiling classification systems used in this study are the best to measure the healthiness of food and beverage purchases across the population? As the authors have noted, grains can be healthy and unhealthy. As such, why is calories per capita from grains a good measure for assessing nutrition-related behaviours? Moreover, the term quality should be replaced with ‘nutritional quality’ throughout the manuscript.

Results:

Line 161-162: Given there is a major focus on rural vs urban disparities in purchasing behaviours, this sentence could clearly describe the extent to which rural households were sampled and included in this study.

Line 172-177: “Top food groups included grains, salty snacks, desserts, mixed dishes and candy as a share of total calories purchased.”

“In comparison, fruits and non-starchy vegetables make up a small percent of calories purchased.”

“Although the top categories of foods and beverages were the same across store type and time, the relative proportion of calories from each group varied widely by store type.”  

As above, in these sentences, it is unclear if these outcomes were measured per household or capita? This information should be consistently reported throughout the manuscript and also added to the table titles/legends.

Line 224-225: “There are few rural-urban differences in the types of foods purchased by low- and high-income groups”.

As previously alluded to, ‘types of foods purchased’ could be replaced by the more scientific outcome measure.

Tables 3-4: The first column requires clear units of measurement across the variables (eg. Fruits, kcal per capita per day?)

Discussion:

Lines 240-242: How did the outcomes vary by urban or rural residence and household income? These initial sentences of the discussion should come back to answering the study aim.

Lines 258: “Top packaged food and beverage categories THAT ARE PURCHASED?”. Some clarification as to whether the authors are referring to purchases (as opposed to market share) is required.

Lines 278-279: Some of the significant differences observed in the results are quite small. Additional discussion of why these differences are meaningful in practical terms (i.e. the real-world) is required. Perhaps it is about contextualising these purchasing practices over longer durations rather than per day?

Lines 292-307: There is some good discussion regarding why differences were observed across store types and the need for retailers to offer healthier food options. Nonetheless, additional consideration of differences in terms of the interactions between pricing, promotional, placement and product (i.e the 4Ps) across store types, could be provided.

Lines 315-316: Additional discussion of the implications of retail interventions for improving health equity could be provided as this appears to be an overarching aim of the study. That is, what can be done at the retail level to reduce nutritional disparities in the US?

Round 2

Reviewer 2 Report

Dear authors,
Thank you for addressing all of my comments and doubts. Now the methodology is clear for me, and the manuscript is well-prepared.